# Augmented Vision-Language Models: A Systematic Review

**Anthony C Davis**                                              *tony.davis@jhuapl.edu*
*Johns Hopkins University*

**Burhan Sadiq**                                                    *bsadiq1@jhu.edu*
*Johns Hopkins University*

**Tianmin Shu**                                                  *tianmin.shu@jhu.edu*
*Johns Hopkins University*

**Chien-Ming Huang**                                        *chienming.huang@jhu.edu*
*Johns Hopkins University*

**Reviewed on OpenReview:** *https://openreview.net/forum?id=DFnPi7Tv6J*

## Abstract

Recent advances in visual-language machine learning models have demonstrated exceptional ability to use natural language and understand visual scenes by training on large, unstructured datasets. However, this training paradigm cannot produce interpretable explanations for its outputs, requires retraining to integrate new information, is highly resource-intensive, and struggles with certain forms of logical reasoning. One promising solution involves integrating neural networks with external symbolic information systems, forming neural symbolic systems that can enhance reasoning and memory abilities. These neural symbolic systems provide more interpretable explanations to their outputs and the capacity to assimilate new information without extensive retraining. Utilizing powerful pre-trained Vision-Language Models (VLMs) as the core neural component, augmented by external systems, offers a pragmatic approach to realizing the benefits of neural-symbolic integration. This systematic literature review aims to categorize techniques through which visual-language understanding can be improved by interacting with external symbolic information systems.

## 1 Introduction

### 1.1 Motivation

Vision-Language Models (VLMs) represent a significant leap forward in artificial intelligence (AI), showing remarkable abilities to interpret complex visual scenes and generate coherent natural language descriptions, powering advancements in tasks such as visual question answering (VQA) (Alayrac et al., 2022) and image/video captioning (Radford et al., 2021). Trained on vast web-scale datasets, these models excel at mapping between visual inputs and textual concepts. However, this end-to-end training paradigm inherently limits their capabilities in several critical ways. VLMs produce outputs without clear justifications, making them difficult to trust or debug without specialized tools (Rudin et al., 2021; Stan et al., 2024). Integrating new factual knowledge or correcting errors typically requires resource-intensive retraining. Furthermore, despite their semantic understanding, VLMs often struggle with tasks that require precise logical deduction, mathematical calculation (for example, accurate object counting), verifiable factual recall of entities within an image, and complex spatial reasoning (Khajuria et al., 2024; Zhang et al., 2025b). These limitations hinder their deployment in high-stakes applications that require precision, reliability, and adaptability. The concept of augmenting VLMs with external information systems has evolved into several distinct paradigms that offer practical solutions to VLM limitations. These augmentation approaches can be broadly categorized by the type of external resource utilized and how it interfaces with the VLM.

*Retrieval-based augmentation* has emerged as one of the most widespread approaches, with Retrieval Augmented Generation (RAG) (Lewis et al., 2021) becoming a common paradigm in both research and commercial applications. These methods retrieve relevant information from external sources to provide context for the VLM's processing. The retrieval mechanisms vary considerably: dense vector-based retrieval uses learned embeddings to find semantically similar content, often employing pretrained encoders like CLIP or custom embedding models; traditional term-based methods like BM25 provide lexical matching capabilities; and structured retrieval from knowledge graphs enables access to explicitly encoded relationships and facts (See Figure 1 for an example of knowledge graph retrieval). While some approaches fine-tune the VLM jointly with the retriever to improve retrieval relevance and integration (Chen et al., 2022b; Rao et al., 2023; Yuan et al., 2023b), many implementations use frozen VLMs with off-the-shelf retrieval systems, demonstrating the flexibility of this augmentation strategy. The retrieved information can be integrated at different stages: as additional input context (prompt augmentation), during the model's reasoning process, or to validate and refine outputs.

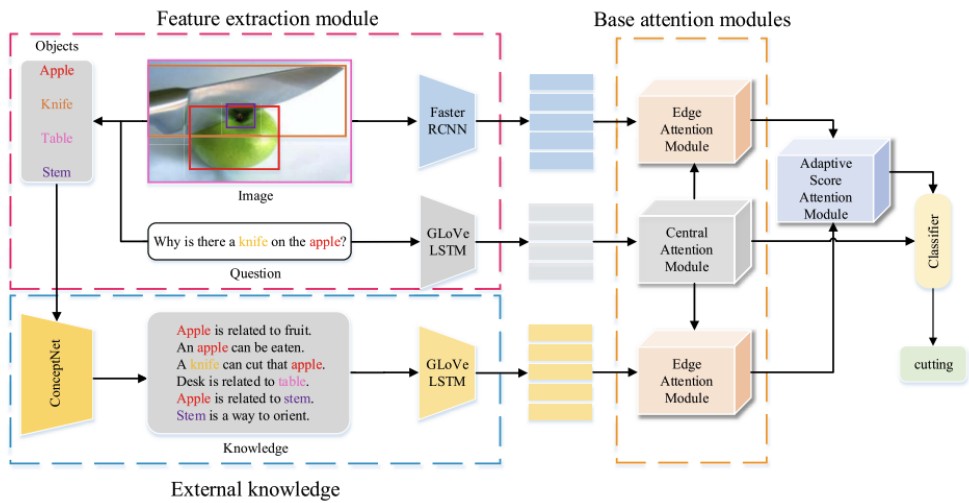

Figure 1: Architecture of the Knowledge-based Augmentation Network (KAN) by Zhang et al. (2020). The system extracts visual features via an object detector module and retrieves external knowledge from ConceptNet with labeled relationships and reliability scores.

*Symbolic computation augmentation* represents another major category where VLMs interface with external computational tools and reasoning engines. Program synthesis approaches enable VLMs to generate executable code (e.g., Python scripts, SQL queries, or domain-specific languages. See figure 2 for an example) that operates on structured representations or queries external systems, with the execution results informing the VLM's outputs. Symbolic reasoning engines such as logic solvers, planning systems, and specialized reasoning frameworks can be invoked to perform precise logical operations that complement the VLM's pattern recognition capabilities. The rapidly evolving paradigm of tool use (Schick et al., 2023; Qin et al., 2023) treats diverse external capabilities (calculators, APIs, specialized vision modules, web browsers) as tools that the VLM can dynamically invoke based on task requirements. Additionally, symbolic graph operations allow VLMs to manipulate structured representations like scene graphs or knowledge graphs through operations such as graph traversal, node matching, or relational reasoning, bridging perceptual understanding with structured symbolic manipulation.

These augmentation strategies offer a pragmatic path forward by building upon the sophisticated visual and language understanding already present in state-of-the-art VLMs, rather than replacing them. Through augmentation, a single VLM can be adapted to diverse tasks without needing to master every capability internally. Importantly, VLMs can be trained when and how to invoke these external resources, discovering effective strategies for combining their internal representations with external capabilities. This approach enables targeted mitigation of specific weaknesses, such as mathematical computation through calculators, factual accuracy through knowledge bases, and complex spatial reasoning through specialized geometric

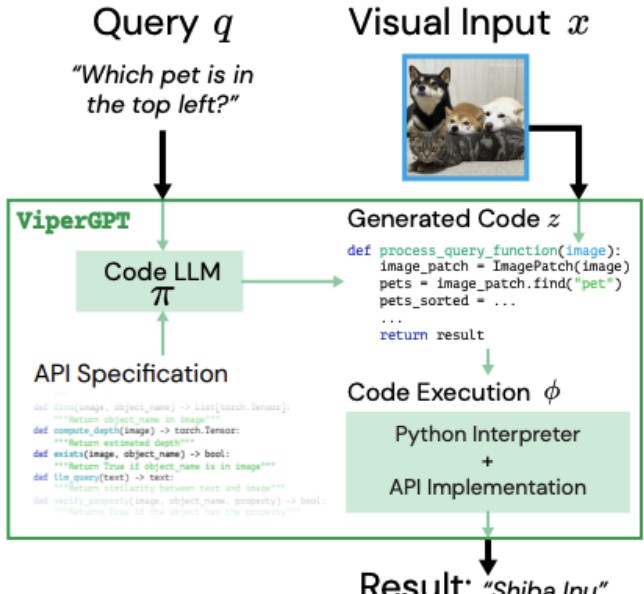

Figure 2: The ViperGPT framework by Surís et al. (2023). Given a visual query and an image/video input, ViperGPT uses a code-generation model (GPT-3 Codex) to generate Python code that composes various vision modules through an API. The generated program makes explicit function calls to vision capabilities (e.g., `find`, `compute_depth`, `count`) and uses Python's built-in logical and mathematical operators to reason about the results.

modules (Surís et al., 2023; Marino et al., 2020; Yi et al., 2018). This systematic review examines how these augmentation techniques are realized in practice, analyzing their implementation strategies, capabilities, and the specific VLM limitations they address.

## 1.2 Augmented Vision-Language Models: Definition and Scope

We define an Augmented Model as a system where external information or computational processes are actively integrated with a neural model's inference operations (before, during, or after its forward pass) to enhance its capabilities. This survey focuses specifically on inference-time augmentation, where external resources are accessed during model execution, not training-time data augmentation. However, the AVLMs we survey may still undergo finetuning to improve their ability to utilize these external resources effectively. This inference augmentation is distinct from prompting techniques like chain-of-thought or in-context examples (Wei et al., 2023; Zhao et al., 2023a), which elicit latent reasoning capabilities without accessing external data or tools.

A Vision-Language Model, for the purpose of this review, is a machine learning model that jointly processes and understands visual and textual modalities, either through generation tasks (e.g., VQA, captioning), alignment tasks (e.g., image-text retrieval, zero-shot classification), or both. This encompasses models that output natural language text as well as those that learn joint representations of vision and language.

Therefore, an AVLM is an VLM integrated with external symbolic information systems, APIs, databases, or other computational tools. An AVLM may involve modifications in VLM neural architecture, or it may involve pre- or post-processing of inputs or outputs of the VLM. Regardless, these integrations aims to overcome the inherent limitations of standalone VLMs and represent a particularly compelling implementation of the augmented neural system concept. A glossary of key terms in this survey is included in Appendix C.

### 1.3 Related Work and Knowledge Gap

The quest to enhance neural models, particularly in the vision-language domain, by incorporating external knowledge or symbolic reasoning has spurred significant research, reflected in several existing surveys. Reviews on knowledge-enhanced multimodal learning (Lymperaiou & Stamou, 2022; Zhao et al., 2023b; Wajid et al., 2023) investigate integrating factual knowledge, often via knowledge graphs or retrieval augmentation, to improve tasks like captioning and VQA. Concurrently, surveys exploring neuro-symbolic approaches (Aditya et al., 2019; Senior et al., 2023; Hitzler et al., 2022; Khan et al., 2024) examine the broader challenge of combining neural perception with symbolic reasoning, often focusing on graph neural networks, spatio-temporal logic, or commonsense knowledge integration for better scene understanding and reasoning. Specific areas like VQA have also been surveyed (Jamshed & Fraz, 2021; Mostafa et al., 2020), tracing the evolution towards models capable of more complex reasoning, sometimes touching upon the need for external knowledge or structured representations.

While these surveys provide valuable context by covering knowledge integration, neuro-symbolic methods, and advances in VQA reasoning, they do not specifically offer a systematic review focused on the augmentation of VLMs through interaction with diverse external symbolic systems and tools. Existing reviews often focus on specific knowledge types (e.g., knowledge graphs) or broader neuro-symbolic theory. There is a knowledge gap in understanding the landscape of techniques specifically designed to connect modern VLMs with external symbolic resources in a flexible, often learned manner (i.e., tool use). Particularly, there is a lack of systematic analysis regarding how these augmentation techniques address core VLM challenges, such as their noted difficulties with precise spatial reasoning (Wang et al., 2024c; Zhang et al., 2025b). Augmentation via external tools or information sources presents a potential pathway to compensate for such weaknesses by providing structured spatial information or enabling interactions with geometric reasoners, at least until VLM architectures intrinsically improve in these areas.

This systematic literature review aims to fill this gap by specifically categorizing and analyzing techniques where VLMs interact with external symbolic information systems or tools to enhance their vision-language understanding capabilities. We seek to provide a structured overview of how these augmentations are implemented, what types of external systems are used, and how they address the limitations of standard VLMs, with a particular interest in emerging tool-use paradigms and their application to challenging visual reasoning tasks.

## 2 Overview: Three Stages of Vision-Language Fusion

The papers surveyed demonstrate a variety of techniques for augmenting vision-language models with external symbolic information systems. The selection of these studies is the result of a systematic literature search conducted according to the PRISMA guidelines (Page et al., 2021), which involved querying academic databases with specific keywords and applying rigorous inclusion/exclusion criteria to identify relevant publications (see Appendix A). This process ensures that the surveyed works specifically target inference-time augmentation and filter out approaches like pure prompting or training-time knowledge integration. To structure this diverse landscape, we categorize the surveyed approaches based on three key characteristics:

- *When* the external interaction occurs relative to the VLM's processing pipeline. We distinguish between Early Fusion (integrating external data at the input stage, influencing initial representations), Middle Fusion (interfacing with external systems during the VLM's internal reasoning or generation steps), and Late Fusion (using the VLM's initial output to trigger external processing, validation, or refinement).

- *What* type of external information or computation is leveraged. This includes Retrieval (accessing pre-existing facts or knowledge from sources like knowledge graphs or text corpora) and Symbolic Computation (generating new information through logical deduction, program execution, or specialized computational tools), or a combination of both.

- *How* the fusion is specifically implemented, detailing the particular mechanisms used in each approach.

This review primarily organizes findings according to the temporal fusion stage (When), as this significantly impacts how external information influences the VLM. Within each temporal category (Early, Middle, Late), we further analyze the type of external interaction (What) and discuss notable implementation details (How). While some sophisticated methods may blend characteristics, this framework provides a structured lens for comparing the underlying principles, capabilities, and trade-offs of different augmentation techniques. The following sections elaborate on the findings for each category, referencing the detailed categorizations presented in the Appendix tables (Tables 2 through 5).

## 3 Early Fusion Methods

Early fusion methods augment the VLM by incorporating external information directly at the input stage, before the core VLM begins its internal processing. This is often the conceptually simplest approach, treating external information as additional context and potentially requiring no VLM architecture changes. Its main advantage is implementation simplicity, offering a direct way to provide context. However, it faces challenges related to the relevance and noise of retrieved information. For example, some implementations use generated image captions as retrieved context which may introduce information loss. The choice between simple prompt augmentation and more structured retrieval encoding depends on the desired level of integration and complexity tolerance. These methods primarily fall into retrieval-based or, less commonly, symbolic computation-based categories, as detailed in Appendix Table 2.

### 3.1 Retrieval-Based Early Fusion

The most common early fusion strategy involves retrieving relevant information from external sources and providing it alongside the primary visual and textual inputs. A primary technique is **Prompt Augmentation**, where retrieved textual context is directly appended to the input prompt, exemplified by Retrieval Augmented Generation (RAG) (Lewis et al., 2021). This retrieved text can originate from various sources. Text/Fact Retrieval draws information from text corpora or knowledge graphs (KGs), using approaches ranging from pre-trained encoders like CLIP without further training to fine-tuning the retriever, possibly jointly with the VLM, for better relevance (see Table 2, column "Retrieval FT"). See Figure 3 for an example of text retrieval using a pretrained vision-language encoder. Reranking retrieved results is often employed to enhance quality (Qu et al., 2024; Liu et al., 2024; Wen et al., 2024). Retrieved KG triplets can also be formatted as text for the prompt (see Table 2, column "Prompt Augmentation"). An alternative form of prompt augmentation uses Image Caption Augmentation, where textual descriptions (captions, labels, Optical Character Recognition (OCR)) are first generated from the visual input, and this text is then used for retrieval or directly added to the prompt, with some methods jointly training the caption generator and retriever (see Table 2, column "Image Caption"). While simplifying the problem to text-based retrieval, this approach risks information loss during captioning.

Instead of appending raw text, another approach uses **Retrieval Encoders** to encode the retrieved information (e.g., KG subgraphs, text passages) into separate embedding vectors. These embeddings then condition the VLM, often through attention mechanisms (Yuan et al., 2023b; Weng et al., 2024; Chen et al., 2022a; Salemi et al., 2023a), Long Short Term Memory models (LSTMs) (Wu et al., 2016), or memory modules (Hu et al., 2022). This allows for a more structured integration of knowledge. Specifically, KG subgraphs can be encoded using Graph Neural Networks (GNNs) (see Table 2, column "Subgraph Enc") or fused with scene graphs (see Table 2, column "KG Conv"). Multimodal KGs can also provide richer representations (Jiang & Meng, 2023).

### 3.2 Symbolic Computation Early Fusion

Integrating the results of symbolic computations at the input stage is rare in the surveyed literature. The primary example identified (Potapov et al., 2019) involves transforming the visual input into a symbolic scene graph. This structured representation, potentially processed by an external symbolic reasoning engine like OpenCog, serves as input or conditioning for the VLM. This approach explicitly introduces symbolic structure early on but depends heavily on robust perception-to-symbol conversion modules.

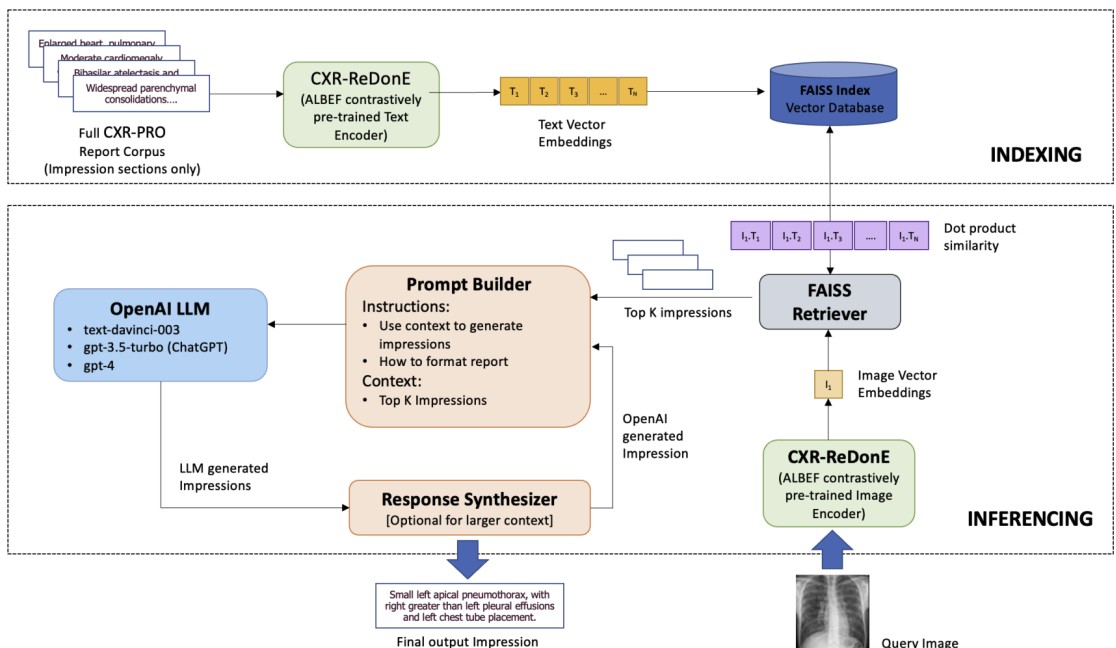

Figure 3: Architecture for Retrieval Augmented Chest X-Ray Report Generation by Ranjit et al. (2023). Text embeddings from radiology impressions are indexed in a vector database. For an input X-ray image, its embedding, generated by a contrastively pretrained vision-language encoder (CXR-ReDonE), is used to retrieve the most similar text (impressions or sentences) from the database. This retrieved text then forms the context for a prompt, along with specific instructions, which is fed to an LLM (e.g., OpenAI GPT models) to generate the final radiology report impression. This process is illustrated for both indexing and inferencing stages.

## 4 Middle Fusion Methods

Middle fusion techniques integrate external information or symbolic computation *during* the VLM's inference stage, allowing interaction with the model's intermediate representations before the final response is generated. Intermediate representations can mean either token-based such as in autoregressive generation, or embedding-based as in dense retrieval techniques. This enables more dynamic and potentially iterative integration compared to early fusion, where external data influences internal processing, reasoning steps, or feature refinement. By allowing external information and symbolic processes to interact with the VLM's internal state, these methods enable context-aware reasoning and iterative refinement. This often involves more complex architectures and training but holds promise for leveraging both neural pattern recognition and symbolic manipulation more effectively. The rise of tool use and agent-based frameworks within this category points towards VLMs acting less as monolithic predictors and more as components in larger reasoning systems, echoing paradigms like Kahneman's System 1 (neural intuition) and System 2 (deliberate symbolic reasoning) (Kahneman, 2011; Booch et al., 2020). These methods, categorized in Appendix Table 3, often involve feedback loops or specialized modules operating alongside main VLM components.

### 4.1 Retrieval-Based Middle Fusion

These methods retrieve external information based on intermediate VLM states and fuse it back into the ongoing computation. One approach is **Dense Retrieval**, which uses dense vector similarity between intermediate VLM representations and a knowledge corpus to find relevant information (often images or text) that is then fused back into the model's layers, typically via attention (Wang et al., 2022b; Lin et al., 2023b; Jia et al., 2023). Another major approach leverages **Graph-Based Retrieval**, primarily using KGs. This includes methods where intermediate visual or textual features trigger **KG Querying**; the retrieved

subgraphs or facts are processed (often with GNNs) and fused with VLM representations, sometimes after extracting visual subgraphs first (see Table 3, column "KG Prompt Augmentation"). Figure 4 illustrates a middle fusion approach that constructs coupled scene and concept graphs, using shared entities as mediums for cross-modal knowledge exchange. Other graph-based methods use **Similarity Measures** between internal VLM representations and KG elements to guide reasoning or weighting, rather than directly injecting KG structure (Wu et al., 2024a; Chae & Kim, 2022; Li et al., 2019; ming Xian et al., 2023; Marino et al., 2020). A significant group focuses on **Concept/Scene Graph Fusion**, explicitly combining internally generated scene graphs with external concept graphs (e.g., from ConceptNet (Speer et al., 2018)), often using GNNs on the combined graphs (see Table 3, column "Concept/Scene Fusion"). More complex structures like **Multimodal KGs** (MMKGs) (Xi et al., 2024; Shi et al., 2022; Santiesteban et al., 2024; Ouyang et al., 2024; Liu et al., 2021) or **Hypergraphs** (Heo et al., 2022; Wang et al., 2024b) are also integrated using specialized graph networks. Finally, **Reinforcement Learning** (RL) can be used to learn policies for querying or integrating external knowledge based on the current state (Bougie et al., 2018).

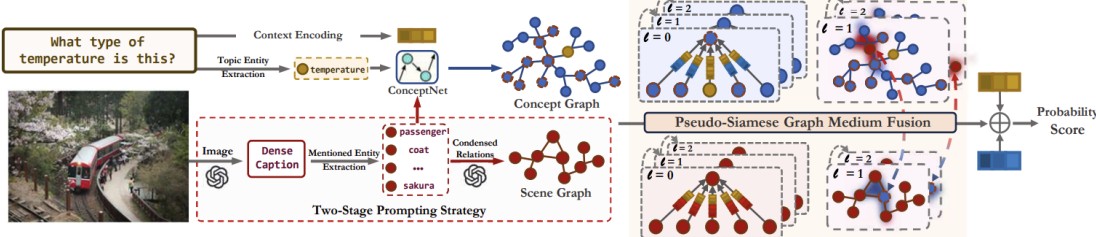

Figure 4: The MAIL (Modality-Aware Integration with LLMs) framework by Dong et al. (2024). The system employs a two-stage prompting strategy: first generating a dense caption through a visual LLM, then constructing a scene graph by extracting spatial and object features as triples (e.g., *(person, wearing, coat)*). These scene graph entities are linked with external knowledge from ConceptNet to form a coupled concept graph containing real-world facts (e.g., *(coat, used_for, warn)*). A pseudo-siamese graph medium fusion module processes both graphs through parallel graph attention networks with different weights, using shared mentioned entities as mediums to enable cross-modal exchange while preserving intra-modal information.

## 4.2 Symbolic Computation Middle Fusion

These methods incorporate symbolic reasoning, calculations, or tool use within the VLM's processing pipeline. One key technique is **Program Synthesis**, where the VLM generates intermediate programs (e.g., functional programs, Python code) operating on symbolic input representations or querying external tools; the execution result influences subsequent VLM processing (Zhang et al., 2022c; 2023e; Hu et al., 2023b), (Shirai et al., 2023, see Figure 5), (Zhang et al., 2023b; Li et al., 2021; Mishra et al., 2024; Xue et al., 2024). Another approach involves integrating **Symbolic Logic Engines**, translating intermediate VLM representations into facts or queries processed by engines like differentiable first-order logic (Zhang et al., 2025a), Answer Set Programming (ASP) (Riley & Sridharan, 2019; Mitchener et al., 2021), Description Logic (Tsatsou et al., 2021), planning domain definition languages (PDDL) (Zhang et al., 2022b; 2023d), temporal logic (Choi et al., 2024), specialized neurosymbolic languages like Scallop (Li et al., 2023d; Huang et al., 2021), or embedding propositional logic operations (Li et al., 2023c). **Vector Symbolic Architectures (VSAs)** represent symbols and perform operations using high-dimensional vectors within the neural architecture (Montone et al., 2017; Kovalev et al., 2021). Some methods perform **Symbolic Graph Operations** directly on graph representations (scene graphs, KGs) during processing, like guided walks or routing (Li et al., 2022c; Liang et al., 2020; Wu et al., 2023; Zhao, 2015; Yang et al., 2020; Zhang et al., 2023f; Hudson & Manning, 2019; Cao et al., 2021). Increasingly popular is **Tool Use**, where the VLM dynamically calls external tools (calculators, APIs, vision algorithms, drawing tools) based on its intermediate state, integrating the tool's output (see Table 3, column "Tool Use"). Lastly, **Self Play** involves using the VLM within a simulated environment where it interacts, uses tools (potentially itself), and learns from feedback (Misiunas et al., 2024).

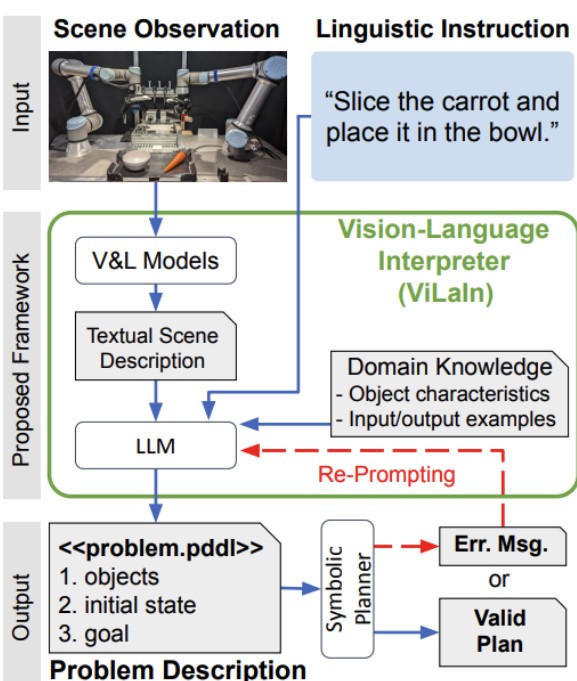

Figure 5: Overview of the ViLaIn approach for VLM planning of robotic actions Shirai et al. (2023). The vision-language interpreter (ViLaIn) generates a problem description from a linguistic instruction and scene observation. The symbolic planner finds an optimal plan from the generated problem description.

### 4.3 Combined Retrieval and Symbolic Computation Middle Fusion

These advanced methods integrate both retrieval and symbolic computation during the forward pass. Many employ **Agent** architectures where the VLM acts as a controller, deciding when to retrieve information and when to use symbolic tools (including sub-agents or code execution) to achieve a goal (see Table 3, column "Agents"). **Other Approaches** combine retrieval (e.g., from ontologies, KGs) with symbolic reasoning (e.g., probabilistic logic, program synthesis, graph walks, concept binding) in bespoke ways for specific tasks like embodied QA, riddle solving, or rumor detection (Besbes et al., 2015; Aditya et al., 2016; Aditya, 2017; Aditya & Baral, 2016; Tan et al., 2021; Liu et al., 2023a; Stammer et al., 2024; Vatashsky & Ullman, 2018; Gao et al., 2023c; 2024).

## 5 Late Fusion Methods

Late fusion methods apply external information retrieval or symbolic computation *after* the VLM has generated an initial output. This external step typically serves to validate, refine, explain, or augment the VLM's output using structured knowledge or precise tools. Late fusion provides a powerful mechanism for verification, refinement, and explanation by applying structured knowledge or precise computations to the VLM's generated output. It leverages the VLM's ability to produce a plausible initial response, which then guides a more targeted external process. This approach is particularly well-suited for enhancing reliability and interpretability, as symbolic steps can act as explicit checks or provide traceable reasoning paths. The main dependency is the quality of the initial VLM output; if it is too vague or incorrect, the subsequent external process may be misguided. These techniques are cataloged in Appendix Table 4.

### 5.1 Retrieval-Based Late Fusion

Here, the VLM's output triggers a targeted retrieval query. In **Dense Retrieval**, the initial VLM output (e.g., answer, rationale) queries a dense retrieval system. The retrieved information (text, facts) is then used

to refine the output or provide supporting evidence (Song et al., 2022a;b; Shi et al., 2024). Alternatively, using **Knowledge Graph Retrieval**, the VLM's output (e.g., generated caption, predicted relationships) queries a KG. Retrieved facts or subgraphs refine the output, for instance, by adjusting probabilities or improving relationship predictions (Gao et al., 2022b; Huang et al., 2020; Xiao & Fu, 2022).

## 5.2 Symbolic Computation Late Fusion

This involves applying symbolic tools or logic engines to the VLM's output. **Program Synthesis** generates programs based on the VLM's output for analysis, validation, or transformation. Examples include generating Python code to verify VQA answers via vision APIs, treating symbolic programs as latent variables, or generating Structured Query Language (SQL) queries from the output (see Table 4, column "Program Synth"). The influential Neural-Symbolic VQA (NS-VQA) approach (Yi et al., 2018), executing programs on scene representations post-prediction, is often adapted. **Symbolic Engines** feed the VLM's output (or derived symbolic representations) into formal logic engines (e.g., Prolog, ASP, Probabilistic Soft Logic) for consistency checking, inference, or validation (Sethuraman et al., 2021; Aditya et al., 2018; Eiter et al., 2022; 2021; Cunnington et al., 2024), or use PDDL for planning (Xu et al., 2022). **Tool Use** involves calling external tools or APIs based on the VLM's output for specialized functions, verification, or generating structured data (Yuan et al., 2023a; Cesista et al., 2024; Cesista, 2024; Zhang, 2023). **Symbolic Graph Operations** perform manipulations on graph representations derived from the VLM's output, such as reasoning over action chains or graph traversals (Li et al., 2023a; Zhan et al., 2021; Saqur & Narasimhan, 2020; Johnston et al., 2023). **Other Approaches** include applying symbolic solvers to latent representations (Singh, 2018), using VLM output confidence to trigger human interaction or further symbolic checks (Bao et al., 2023), or updating conversational memory based on the response (Verheyen et al., 2023).

## 5.3 Combined Retrieval and Symbolic Computation Late Fusion

These methods combine both retrieval and symbolic computation after the initial VLM output. Typically, the VLM output is parsed into a logical form, relevant domain knowledge (facts or programs) is retrieved, and a symbolic reasoner (e.g., probabilistic logic, ASP) derives the final answer (Sachan, 2020; Basu et al., 2020). The AQuA framework (Basu et al., 2020) is depicted in Figure 6.

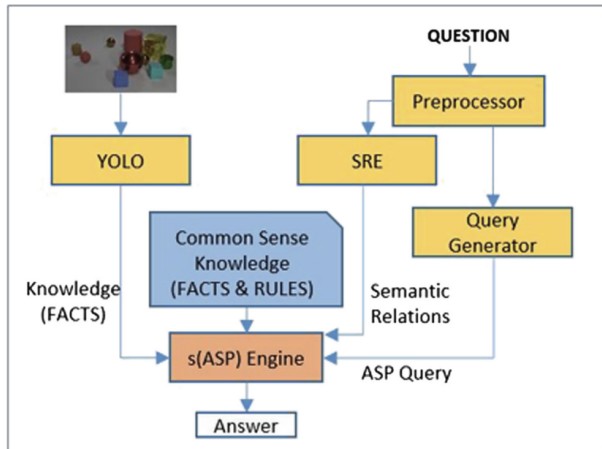

Figure 6: The architecture of the AQuA framework Basu et al. (2020). It consists of five main modules: (i) YOLO for object detection and feature extraction, (ii) a Preprocessor for the natural language question, (iii) a Semantic Relation Extractor (SRE), (iv) a Query Generator based on semantic analysis, and (v) a retrieval based Commonsense Knowledge module leveraging. The system utilizes an ASP engine for symbolic reasoning.

# 6 Discussion

The studies reviewed in this paper underscore the growing success of incorporating external symbolic information into vision-language models across different fusion paradigms. Here we discuss key observations, challenges, and potential directions for future research stemming from these findings.

## 6.1 Domain-Specific Limitations and Augmentation Solutions

The integration of external symbolic systems in vision-language models addresses fundamental perceptual and reasoning limitations that distinguish visual understanding from purely textual tasks. We organize domain-centric limitations of VLMs along five phenomena that recur across the surveyed studies: spatial, temporal, knowledge grounding, physical commonsense, and action/embodiment (STKPA). Each category of limitation has driven the development of specific augmentation patterns that leverage the complementary strengths of neural perception and symbolic computation. Orthogonal to this axis is how external structure is injected, via retrieval, program/tool execution, or logic/constraint checking, and when it is injected (early, middle, or late fusion; see Sections 3– 5). The matrix in Table 1 summarizes typical augmentation patterns observed for each phenomenon. Across the 264 papers in our corpus, each work is aligned with at least one STKPA category, as shown in Tables 2 to 4. We will discuss each domain and highlight some example augmentation solutions and their associated datasets.

### 6.1.1 Spatial numeracy and geometry

VLMs struggle with tasks requiring exact measurements or counts within visual scenes, with error rates exceeding 30% on simple counting tasks (Wang et al., 2024c). This limitation stems from the continuous nature of visual features in neural representations conflicting with discrete spatial reasoning requirements.

Effective augmentation systems externalize discrete structure through two primary families of approaches. First, program synthesis methods compile questions to programs that operate on structured scene representations. Neural-Symbolic VQA (Yi et al., 2018) achieved 99.8% accuracy on CLEVR through explicit program execution over scene graph representations, effectively saturating the benchmark with a 15+ percentage point improvement over end-to-end neural baselines. Second, visual API orchestration methods like ViperGPT (Surís et al., 2023) generate Python code that composes heterogeneous computer vision modules such as detectors, object/semantic segmenters, or depth estimators, each providing complementary symbolic information about the visual scene. This compositional approach allows VLMs to leverage specialized perception modules' strengths while maintaining flexibility through learned orchestration policies.

Spatial reasoning benchmarks have been fundamental in evaluating AVLMs' ability to understand geometric relationships and counting (Table 5). The CLEVR family of datasets (Johnson et al., 2016) provides synthetic but precisely controlled evaluation of spatial reasoning, with extensions like Super-CLEVR (Li et al., 2022e) testing domain robustness and CLEVR-POC (Abraham et al., 2024) introducing partial observability challenges. Scene graph datasets including Visual Genome (Krishna et al., 2016) and VCD (Shen et al., 2024) evaluate models on real-world spatial relationships between objects. More recent benchmarks like SOK-Bench (Wang et al., 2024a) combine spatial and knowledge requirements, while specialized datasets test whether VLMs can reason about times and locations (Zhang et al., 2023a) or answer complex visual information-seeking questions (Chen et al., 2023).

### 6.1.2 Temporal and causal ordering

In video understanding, VLMs frequently misinterpret temporal sequences and causal relationships, with accuracy drops of 20-40% on tasks requiring temporal ordering compared to static image reasoning (Yi et al., 2019). The absence of explicit temporal reasoning mechanisms in standard architectures necessitates augmentation with structured temporal representations.

Middle-fusion pipelines address this by tracking entities and actions through specialized tools (detection, tracking, pose estimation, optical flow) to construct event sequences that programs or planners can reason over. Logic-based approaches apply temporal operators and constraint checks to predicted timelines, with

| | Retrieval | Symbolic Computation |
|---|---|---|
| **S**patial | KG fact retrieval to bias relations/attributes (Marino et al., 2020; Li et al., 2020); dense region/sentence retrieval for spatial cues (Chen et al., 2022b; Lin et al., 2022); graph-based retrieval of related objects/relations (Wang et al., 2022c; Zhu et al., 2020b) | Executable visual programs over scene graphs (NS-VQA) (Yi et al., 2018); Python tool orchestration for counting/geometry (detector/segmenter/depth) (Surís et al., 2023); visual program distillation (Hu et al., 2023b); ASP/logic verification of spatial predicates (Basu et al., 2020; Eiter et al., 2022; Sethuraman et al., 2021); probabilistic neural–symbolic constraints (Vedantam et al., 2019) |
| **T**emporal | Knowledge-guided caption/entity retrieval for event context (Xi et al., 2024; Hou et al., 2020); ontology lookup of task/event relations (Jiang et al., 2020) | LLM-orchestrated video tools (tracking, OCR/ASR, shot segmentation) (Fan et al., 2024); program traces over events (Zhang et al., 2023e); symbolic activity reasoning with tool-augmented execution (Wu et al., 2023); temporal logic/order constraints and ASP checks (Choi et al., 2024; Eiter et al., 2022) |
| **K**nowledge grounding | Outside-knowledge retrieval (text/KG, reranking) for OK-VQA (Marino et al., 2019; Chen et al., 2022b; Wen et al., 2024); multi-source multimodal retrieval (entities, captions, KG) (Zhu et al., 2020b; Salemi et al., 2023a; Gui et al., 2021) | Code/program generation to query KGs/APIs or compose operators (Subramanian et al., 2023; Zhang, 2023); agentic tool-use for information seeking (Hu et al., 2023c); PSL/formal-logic consistency and entailment (Aditya et al., 2016; Sethuraman et al., 2021; Eiter et al., 2022) |
| **P**hysical common-sense | Affordance/functional KB retrieval (e.g., ConceptNet) (Speer et al., 2018; Marino et al., 2020); knowledge-aided captioning (Huang et al., 2020) | Differentiable/explicit physics for feasibility (Huang et al., 2023); vision-tool composition for metric-/geometry checks (Surís et al., 2023); 3D symbolic grounding of objects/relations (Hsu et al., 2023); rule/constraint checking for stability, containment, and causal function (Aditya et al., 2018; Vedantam et al., 2019; Zhao, 2015) |
| **A**ction / embodiment | External maps/knowledge for navigation and planning (Li et al., 2022d; Ni et al., 2023); VL-action models transferring web knowledge for control (Brohan et al., 2023); robotic manipulation knowledge bases (Gao et al., 2023b) | Planner interfaces (PDDL/ASP) invoked from VLM outputs (Shirai et al., 2023; Zhang et al., 2022b); plan–execute–observe loops grounded by VLMs (Zhang et al., 2023b; Gao et al., 2023b); tool documentation for zero-shot tool use (Hsieh et al., 2023); planner validity/safety constraints and neuro-symbolic agent frameworks (Xu et al., 2022; Cunnington et al., 2024) |

Table 1: Reasoning domains and associated augmentation mechanisms with example implementations

temporal logic systems (Choi et al., 2024) showing 15-25% improvements on temporal ordering tasks. Retrieval of script knowledge or prototypical event progressions biases hypotheses toward plausible sequences, while late-fusion verifiers test temporal feasibility against these priors. These augmented approaches separate perceptual feature extraction from logical temporal inference, allowing each component to operate in its optimal representational space.

Temporal reasoning evaluation is primarily anchored by CLEVRER (Yi et al., 2019) (Table 5), which extends spatial reasoning into the temporal domain through collision events and causal chains in synthetic videos. The Compositional 4D dataset (Wang et al., 2024d) further challenges models with four-dimensional scene understanding requiring both temporal and physical reasoning. While fewer dedicated temporal datasets exist compared to other domains, temporal reasoning is often implicitly tested in embodied agent benchmarks and video understanding tasks that require tracking state changes over time.

### 6.1.3 Knowledge-intensive grounding

The challenge of linking language to specific visual entities requiring external knowledge (identifying landmarks, recognizing famous individuals, understanding fine-grained categories) represents a fundamental limitation where learned pattern matching proves insufficient due to sparse training data for highly specific instances (Kalai et al., 2025).

Multimodal knowledge graph navigation addresses this through explicit retrieval and graph-based reasoning. VQA-GNN (Wang et al., 2022c) demonstrated 4.6% improvement on GQA by coupling internally generated scene graphs with external concept graphs through graph neural networks that enable message passing between visual features and symbolic knowledge. KAT (Gui et al., 2021) achieved 53.1% on OK-VQA, a 5.1% improvement over PICa which relied solely on GPT-3's parametric knowledge (Yang et al., 2022). The integration of structured knowledge graphs like ConceptNet enables more accurate entity recognition through explicit retrieval of facts about identity, function, and long-tail categories, rather than relying solely on learned pattern matching.

Knowledge-based VQA represents the largest category of benchmarks for AVLMs (Table 5). The progression from general VQA (Agrawal et al., 2015) to fact-based FVQA (Wang et al., 2016; Lin et al., 2023c) and knowledge-aware KVQA (Shah et al., 2019) reflects increasing demands for external knowledge. OK-VQA (Marino et al., 2019) and its successor (Reichman et al., 2023) have become standard benchmarks requiring knowledge beyond visual content. Specialized variants target encyclopedic knowledge (Mensink et al., 2023), cultural domains (Agarwal et al., 2024), named entities (Lerner et al., 2022; Qiu et al., 2024), and synthetic knowledge generation (Su et al., 2024). Advanced reasoning benchmarks like A-OKVQA (Schwenk et al., 2022) and Visual Riddles (Bitton-Guetta et al., 2024) combine knowledge requirements with complex compositional reasoning, while CRIC (Gao et al., 2019) tests compositional reasoning on vision and commonsense.

### 6.1.4 Physical commonsense and affordances

VLMs often lack robust understanding of physical properties and object affordances visible in scenes, commonly misunderstanding material properties, physical constraints, or predicted behavior under manipulation (Yi et al., 2019; Wang et al., 2024d). This gap necessitates specialized bridging mechanisms that translate between continuous visual features and discrete symbolic representations.

Two augmentation routes have proven effective. First, symbolic vector or graph representations bridge continuous perception and discrete physical predicates. Vector Symbolic Architectures (Montone et al., 2017; Kovalev et al., 2021) represent spatial relationships as high-dimensional vectors supporting symbolic operations like binding and unbinding within the neural architecture itself. Graph neural networks operating on coupled scene-concept graphs (Dong et al., 2024; Song et al., 2023) enable cross-modal knowledge exchange, with ConceptNet integration showing 8-12% improvements on tasks requiring physical commonsense reasoning. Second, tool-enabled pipelines call physics or geometry modules (depth estimation, meshing, simple simulators) to test feasibility or measure quantities before answering. The structured nature of external knowledge allows models to access explicit relationships like "glass IsA fragile" or "liquid HasProperty flows_downward" that may not be reliably encoded in learned parameters.

Physical reasoning evaluation is less explicitly represented in current benchmarks but appears implicitly across multiple datasets (Table 5). The Compositional 4D dataset (Wang et al., 2024d) specifically targets understanding of physical dynamics and material properties across time. Visual Commonsense Reasoning (VCR) (Zellers et al., 2018) requires understanding of physical plausibility and social dynamics, while explainable reasoning benchmarks (Cao et al., 2019) often involve physical world knowledge. The relative scarcity of dedicated physical commonsense benchmarks represents a gap in current evaluation frameworks, despite this being a critical limitation of VLMs.

### 6.1.5 Action and embodiment

In robotics and embodied AI applications, the challenge of translating visual scenes and natural language instructions into executable action sequences requires precision and verifiability that standalone VLMs cannot

provide. Augmentation patterns treat VLMs as interpreters that generate symbolic action specifications for downstream execution.

Methods like ViLaIn (Shirai et al., 2023) generate PDDL specifications that symbolic planners execute with guaranteed optimality properties. Systems employing Answer Set Programming (Zhang et al., 2022b) or other formal planning languages leverage logical consistency checking. Many approaches implement plan–execute–observe–reflect loops that re-plan on mismatch, with retrieval supplying action/operator libraries and environment maps. Tool interfaces expose planners and low-level skills through well-defined APIs, while safety and validity are enforced through logic layers checking preconditions, effects, and constraints. This vision-to-symbol translation separates flexible perception from rigorous planning, enabling reliable task execution in physical environments.

Embodied AI benchmarks evaluate AVLMs' ability to translate understanding into executable actions (Table 5). Web-based environments like WebArena (Zhou et al., 2023) and ScreenAgent (Niu et al., 2024) test agents on realistic computer control tasks, while Spider2-V (Cao et al., 2024) focuses on data science workflows. Robotic manipulation benchmarks (Gao et al., 2023b) evaluate physical grounding and tool use in real-world settings. These agent-focused evaluations represent a shift from passive question-answering to active, goal-directed interaction with environments, requiring integration of perception, reasoning, and action execution.

## 6.2 Common Architectural Patterns in AVLMs

While the surveyed papers employ diverse implementation strategies, our analysis reveals three fundamental architectural patterns that have emerged as dominant paradigms for augmenting vision-language models. These patterns represent crystallized design solutions that address specific computational challenges in vision-language understanding. These patterns distribute differently across fusion stages, with retrieval-based approaches primarily operating at the input stage, while symbolic computation patterns span both middle and late fusion paradigms.

### 6.2.1 The Retrieval-Integration Pipeline Pattern

The retrieval-reasoning pipeline has evolved from simple keyword matching to sophisticated neural retrieval systems. The evolution of this pattern over the past decade (2016-2024) reflects increasing sophistication in how external knowledge is indexed, accessed and integrated in the following ways:

- **Early Simple Retrieval (2016-2018)**: Direct keyword matching or TF-IDF based retrieval from knowledge bases, with retrieved facts concatenated to prompts (Wang et al., 2015; Narasimhan & Schwing, 2018)

- **Learned Dense Retrieval (2019-2020)**: Introduction of learned embeddings using pretrained encoders like CLIP, enabling semantic similarity search (Li et al., 2020; Zhang et al., 2020)

- **Joint VLM-Retriever Training (2021-2022)**: End-to-end training of retriever and VLM components, optimizing retrieval for downstream task performance (Chen et al., 2022b; Gui et al., 2021)

- **Tool-Augmented Retrieval (2023-2024)**: Retrieval as one tool among many, with VLMs learning when and what to retrieve (Yan & Xie, 2024; Hao et al., 2024b)

Critical design decisions in this pattern include the choice of embedding space, where CLIP-based retrieval demonstrates strong performance on vision-language tasks (Gui et al., 2021), retrieval granularity with sentence-level retrieval showing advantages for factoid questions (Chen et al., 2022b), and integration mechanism where attention-based fusion consistently outperforms simple concatenation (Yuan et al., 2023b; Weng et al., 2024).

### 6.2.2 The Intermediary Program Pattern

This pattern treats program synthesis as a bridge between neural perception and symbolic computation, with the VLM generating executable code that operates on structured representations or invokes external

tools. The generated programs serve as interpretable reasoning chains that can be verified, debugged, and modified. Importantly, we distinguish between domain-specific program synthesis and general-purpose code generation, as they represent different points on the expressiveness-tractability spectrum.

This pattern's evolution has followed a similar progression to the retrieval-reasoning pipeline, moving from static patterns to dynamic, learned behaviors:

- **Static Logic on VLM Outputs (2018-2019)**: Early approaches applied predefined symbolic logic engines (ASP, Prolog) to VLM-extracted scene graphs, requiring manual rule specification (Aditya et al., 2018; Riley & Sridharan, 2019)

- **Domain-Specific Program Synthesis (2020-2022)**: VLMs learned to generate programs in constrained domain-specific languages (DSLs) with guaranteed executability. NS-VQA (Yi et al., 2018) synthesizes functional programs over a fixed set of visual primitives, while Zhang et al. (2025a) generates first-order logic expressions

- **General-Purpose Code Generation (2022-2023)**: Shift to generating Python or SQL code with broader expressiveness but without execution guarantees. ViperGPT (Surís et al., 2023) generates unrestricted Python code composing vision APIs, while Gupta & Kembhavi (2022) produces Python programs for visual reasoning

- **Dynamic Tool Orchestration (2023-2024)**: VLMs as orchestrators selecting and composing heterogeneous tools including APIs, specialized models, and code execution environments (Hu et al., 2023c; Wu et al., 2024b; Liu et al., 2023b)

The key architectural components include: (1) Program specification language - DSLs offer tractability with limited expressiveness (NS-VQA's 20 primitives achieve 99.8% on CLEVR (Yi et al., 2018)), while general-purpose languages enable broader capabilities but require error handling; (2) Execution environment - ranging from symbolic executors for DSLs to sandboxed Python interpreters with vision library access; (3) Error handling mechanisms - recent approaches like Mishra et al. (2024) incorporate execution feedback for iterative refinement. ViperGPT demonstrates the general-purpose approach's flexibility, achieving strong performance across diverse visual reasoning tasks through unrestricted Python generation (Surís et al., 2023).

### 6.2.3 The Graph Fusion Pattern

Graph-mediated fusion explicitly models relationships between visual elements and external knowledge through graph structures, enabling structured reasoning over combined perceptual and symbolic information. This pattern typically involves three stages: graph construction (from visual input and/or external knowledge), graph alignment (connecting visual and symbolic graphs), and graph neural network processing for joint reasoning.

**Key Design Variations:**

- **Scene-Concept Graph Coupling**: Methods like VQA-GNN (Wang et al., 2022c) and MAIL (Dong et al., 2024) construct parallel scene graphs (from images) and concept graphs (from knowledge bases), using shared entities as bridges. Graph attention networks enable cross-modal message passing while preserving intra-modal structure, with VQA-GNN showing 4.6% improvement on GQA through this approach (Wang et al., 2022c)

- **Multimodal Knowledge Graph Integration**: Approaches incorporating MMKGs (Xi et al., 2024; Liu et al., 2021) where nodes contain both visual exemplars and textual descriptions, enabling richer cross-modal grounding through joint embedding spaces

- **Dynamic Graph Construction**: Methods that construct query-specific subgraphs rather than using fixed graph structures, with Li et al. (2022b) demonstrating improved efficiency through adaptive graph pruning.

Critical implementation choices include graph representation (heterogeneous vs. homogeneous nodes), alignment mechanisms (entity matching vs. learned attention), and message passing strategies (synchronous vs. asynchronous updates). The graph-mediated pattern provides a principled way to preserve structural information while enabling neural reasoning, making it particularly effective for tasks requiring explicit relational understanding between visual elements and conceptual knowledge.

### 6.2.4 Cross-Pattern Observations

Several key insights emerge from analyzing these patterns:

1. **Temporal Distribution**: Retrieval-based approaches concentrate in early fusion (where they modify inputs), while program and graph-based patterns distribute across middle and late fusion stages (where they can interact with intermediate representations or refine outputs).

2. **Complementarity**: The most successful recent systems combine patterns - for example, using retrieval to gather relevant facts, then applying program synthesis for precise computation over retrieved information (Castrejon et al., 2024; Lu et al., 2023).

3. **Interpretability-Simplicity Trade-off**: Program-based approaches offer highest interpretability through executable traces but require more specialized, task-specific training. Graph-based methods provide moderate interpretability through explicit relational structure. Pure retrieval offers limited interpretability but is simplest to implement.

4. **Scalability Characteristics**: Retrieval scales well with knowledge base size using approximate nearest neighbor search (Chen et al., 2022b), program synthesis complexity grows with the size of the DSL or API set (Yi et al., 2018), while graph methods face quadratic complexity in number of nodes, requiring approximation techniques for large graphs (Wang et al., 2022c).

These architectural patterns provide a technical foundation for designing augmented vision-language systems, with the choice of pattern depending on task requirements for accuracy, interpretability, and computational efficiency.

### 6.3 Future Directions for Vision-Centric Augmentation Research

**Cross-Modal Knowledge Integration Through Bidirectional Grounding**: Rather than treating retrieved information as passive context, future research should explore augmentation techniques that leverage cross-modal correspondences more deeply. For example, retrieved knowledge about typical spatial layouts (e.g., "monitors are usually on desks") could constrain visual parsing, while visual evidence could trigger targeted knowledge retrieval. Multimodal knowledge graphs where nodes represent visual concepts with both image exemplars and textual descriptions (Jiang & Meng, 2023; Xi et al., 2024) could enable richer cross-modal reasoning through joint embedding spaces. This bidirectional interaction between symbolic knowledge and visual processing could improve both grounding accuracy and reasoning efficiency by ensuring consistency between what the model sees and what external knowledge suggests should be present.

**Embodied Vision-Language Systems with Reinforcement Learning for Tool Use**: The integration of VLMs with physical embodiment and interactive environments presents unique augmentation opportunities beyond one-shot prediction. Embodied systems can iteratively refine understanding through interaction: moving cameras to new viewpoints, manipulating objects to reveal hidden properties, or executing actions to test hypotheses about the physical world (Gao et al., 2023b). Recent advances in reinforcement learning for visual tool use, exemplified by approaches like VTool-R1 (Wu et al., 2025), demonstrate how RL can train VLMs to dynamically select and compose vision tools based on environmental feedback and task requirements. Future research should develop reward functions that balance exploration (trying new tool combinations) with exploitation (using known effective strategies), while incorporating human preferences to ensure safe and interpretable tool usage patterns. This includes learning when to request human intervention for ambiguous visual scenes or safety-critical decisions in autonomous systems.

**Interpretable Visual Program Synthesis for Safety-Critical Applications**: While current visual programming approaches demonstrate success on controlled benchmarks (Surís et al., 2023; Gupta & Kembhavi, 2022), deploying these systems in safety-critical domains requires advances that combine program synthesis with interpretability mechanisms. Future systems must handle partial observability through techniques like uncertainty-aware program generation that explicitly represents confidence in different execution paths. (Bao et al., 2023; Vedantam et al., 2019; Chae & Kim, 2022) The challenge of providing interpretability becomes paramount in high-stakes applications. Systems should produce visual reasoning chains that show which image regions were examined, what external tools were invoked, and how intermediate results combined to reach conclusions. This requires developing visualization techniques that render program execution traces overlaid on images with attention heatmaps, tool call annotations, and confidence scores. For medical imaging, autonomous driving, or industrial inspection, these interpretable execution traces serve dual purposes, enabling domain experts to verify correctness and providing auditable records for regulatory compliance. The key is balancing completeness (showing all reasoning steps for full transparency) with comprehensibility (avoiding overwhelming users with excessive detail through hierarchical or interactive visualization approaches).

**Scalable Visual Program Libraries and Transfer Learning**: Current visual programming approaches often require task-specific program templates or limited APIs. Future research should develop methods for building compositional program libraries that grow through experience, enabling VLMs to synthesize increasingly complex vision pipelines by combining learned subroutines. This includes meta-learning approaches that discover reusable visual reasoning patterns across tasks and transfer learning techniques that adapt programs from source domains (where supervision is abundant) to target domains (where it is scarce). The vision-language community would benefit from standardized APIs and benchmark tasks for visual program synthesis, analogous to how HuggingFace standardized model interfaces for NLP.

**Adaptive Augmentation Based on Task Uncertainty**: Rather than applying fixed augmentation strategies, future AVLMs should dynamically determine when and how to invoke external resources based on uncertainty estimation. For instance, a model confident in its counting ability for sparse scenes might bypass external tools, while requesting symbolic computation for cluttered environments. This adaptive approach requires developing calibrated uncertainty measures for different aspects of visual reasoning (spatial relationships, object recognition, attribute prediction) and learning policies that optimize the trade-off between accuracy gains and computational costs of augmentation.

## 7 Conclusion

Vision-Language Models have revolutionized AI's ability to connect vision and language, yet standalone models struggle with factual accuracy, complex reasoning, adaptability, and interpretability. This systematic review charted the landscape of Augmented Vision-Language Models (AVLMs), which overcome these limitations by integrating VLMs with external symbolic information systems and computational tools. We surveyed a diverse range of techniques, categorizing them by fusion timing (early, middle, late) and the nature of augmentation (retrieval, symbolic computation, combined), revealing a clear consensus: augmenting VLMs significantly boosts performance and interpretability on knowledge-intensive and reasoning-heavy tasks by synergizing neural pattern recognition with symbolic precision (Marino et al., 2020; Vedantam et al., 2019; Bitton-Guetta et al., 2024; Yan & Xie, 2024; Hu et al., 2023c). A particularly powerful paradigm emerging from this landscape is tool use, which offers a flexible and unifying abstraction for AVLM design. This approach frames the VLM as an intelligent orchestrator trained to select and utilize external capabilities (such as knowledge bases, calculators, code execution, specialized algorithms, formal reasoners) encapsulated as "tools," enabling modularity and scalability. Significant challenges remain in managing interaction complexity, ensuring scalability and efficiency, guaranteeing robustness against unreliable external inputs, developing comprehensive evaluation methods, and refining the tool integration mechanisms themselves. Nevertheless, the advancement of AVLMs, particularly through the lens of tool use, represents a crucial progression towards more capable, reliable, and trustworthy AI systems that effectively blend neural perception with symbolic reasoning, allowing them to not only see and describe the world but also reason about it with greater depth, accuracy, and transparency.

# A    Methodology

This section describes the process of gathering relevant articles for this survey, following the Preferred Reporting Items for Systematic reviews and Meta-Analyses (PRISMA) guidelines. The goal of this approach is to avoid bias when selecting what papers to review, focusing on the merits of the paper and the relevancy to the topic of AVLMs.

## A.1    Search Strategy

### A.1.1    Databases and Search Queries

We utilized two primary databases for our literature search:

- **Google Scholar**: Known for its extensive coverage of scholarly publications across disciplines.

- **Semantic Scholar**: Provides advanced search capabilities and citation analysis, facilitating the identification of semantically relevant works.

### A.1.2    Search Terms

We formulated specific search queries to capture studies related to augmented vision-language models interacting with symbolic systems during inference. The search strategy used the strengths of both databases by employing an iterative process of testing and refining the search query until the resulting set of papers was adequately relevant. Google Scholar is more sensitive to the inclusion of keywords, and so we used a combination of Boolean operators to refine the results effectively.

The search query used in Google Scholar was:

```
"("augmented" OR "knowledge" OR "knowledge graphs" OR
"knowledge augmentation" OR "commonsense knowledge" OR
"commonsense reasoning" OR "tool use" OR
"retrieval augmented" OR "retrieval-augmented" OR
"external knowledge" OR "neural symbolic" OR
"neural-symbolic" OR "symbolic")
AND
("vision-language" OR "vision language" OR
"visual question answering" OR "image question answering" OR
"video question answering" OR "image caption" OR
"video caption" OR "image text" OR "spatial reasoning" OR
"visual reasoning")
AND
("neural network" OR "machine learning" OR
"artificial intelligence" OR "deep learning")
-"virtual reality" -"augmented reality"
```

Semantic Scholar is less sensitive to keywords and more of a semantic search, so for this database, we employed a set of targeted queries to capture key aspects of our research focus:

- "Commonsense reasoning in visual question answering"

- "Knowledge graphs for image or video captioning"

- "External knowledge in visual reasoning"

- "Neural-symbolic vision-language models"

- "Tool use in vision-language tasks"

- "Retrieval-augmented image question answering"

- "Symbolic reasoning in AI for vision"

- "Commonsense in image-text models"

- "Neural-symbolic visual question answering"

- "Multimodal knowledge graph LLM"

## A.2 Inclusion and Exclusion Criteria

To ensure the relevance and quality of the studies included in this review, we established clear inclusion and exclusion criteria.

### A.2.1 Inclusion Criteria

- **Relevance**: Studies that describe machine learning models integrating external symbolic information systems during inference.

- **Language**: Publications written in English.

- **Implementation Focus**: Papers providing detailed descriptions of implementation methods rather than purely conceptual or theoretical discussions.

- **Vision-Language Tasks**: Research focusing on tasks such as visual question answering, image captioning, and video captioning where the input is imagery and/or text and the output is natural language text.

### A.2.2 Exclusion Criteria

We excluded studies that did not align with the focus of this review, such as:

- **Prompting Techniques**: Research solely on prompt engineering or techniques that rely on internal reasoning patterns without external data augmentation (e.g., chain-of-thought prompting).

- **Self-Prompting/Recursive Prompting**: Methods that involve iterative querying without integration of external symbolic information systems.

- **Synthetic Data Generation**: Studies focusing on generating synthetic data to improve model performance without external symbolic system interaction.

- **Architectural Modifications Without External Integration**: Papers discussing model architectures like vision encoder adapters for large language models that do not involve external symbolic systems during inference.

- **Training with Structured Knowledge**: Research that involves training models with external knowledge but does not allow for the external knowledge to be modified or read during inference (e.g., methods where external knowledge is embedded in model parameters).

## A.3 Selection Process

The selection process involved several iterative steps to refine and identify the most relevant studies.

### A.3.1 Initial Search Results

- **Google Scholar**: The search yielded **980 papers** after filtering by category and removing irrelevant results based on titles and abstracts.

- **Semantic Scholar**: The targeted queries returned **1,332 papers**.

### A.3.2   Total Papers Collected

In total, **2,312 papers** were collected from both databases.

### A.3.3   Relevance Scoring

In alignment with the theme of augmented models, we utilized the **GPT-4o OpenAI model (gpt-4o-2024-08-06)** to assist in the relevance assessment:

- **Automated Categorization**: GPT-4o was prompted to categorize each paper and assign a relevance score ranging from 1 to 10 based on the alignment with the review topic.

- **Threshold for Inclusion**: Papers scoring less than **8 out of 10** were excluded from further consideration.

- **Iteration and Validation**: The relevance scoring process was iterated, and we ensured that all highly relevant papers were retained, even if they narrowly missed the initial threshold.

### A.3.4   Manual Screening

- **Total Papers After Automated Filtering**: **616 papers** remained after applying the relevance threshold.

- **Full-Text Assessment**: We conducted a thorough manual review of the full text of these papers.

- **Final Selection**: After removing duplicates and papers not meeting the inclusion criteria, **264 papers** were selected for detailed analysis. See Figure 7.

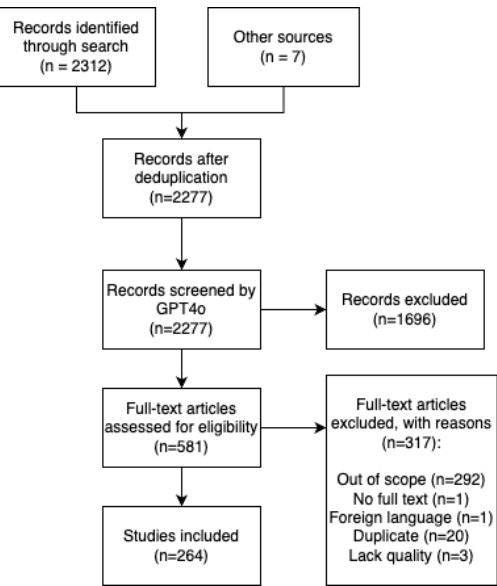

Figure 7: PRISMA Flowchart

### A.4   Data Extraction and Synthesis

From the selected studies, we extracted pertinent information to facilitate a comprehensive understanding of the methods:

- **Integration Techniques**: Description of how external systems were integrated with vision-language models, classified into early fusion, middle fusion, and late fusion methods.

- **Types of External Symbolic Systems**: Categorization of external symbolic systems used, such as knowledge graphs, symbolic logic engines, and program synthesis tools.

- **Tasks Addressed**: Identification of the specific vision-language tasks tackled by each study, including visual question answering, image captioning, and others.

- **Implementation Details**: Detailed examination of the models' architectures, including the interaction mechanisms with external symbolic systems during inference.

### A.5 Quality Assessment

We assessed the quality of the included studies based on:

- **Clarity of Methodology**: Transparency and reproducibility of the methods described.

- **Experimental Rigour**: Adequacy of experimental design, including dataset usage, evaluation protocols, and statistical significance of results.

- **Contribution to the Field**: The extent to which the study advanced understanding or provided innovative solutions in augmented vision-language models.

### A.6 Limitations

While we aimed for a comprehensive review, certain limitations exist:

- **Publication Bias**: Unpublished works or those not indexed in the selected databases may have been missed.

- **Language Restriction**: Non-English publications were excluded, which may omit relevant research conducted in other languages.

- **Dynamic Field**: Given the rapidly evolving nature of machine learning research, new studies may have emerged after the completion of our search.

- **AI Bias**: The use of GPT4o in filtering of papers could potentially remove relevant search results.

By following this systematic approach, we ensured a thorough and unbiased selection of relevant literature, providing a solid foundation for the subsequent analysis and discussion in this review.

## B Categorization Tables

This section contains the tables categorizing the surveyed papers based on the fusion method (Early, Middle, Late) and the type of augmentation (Retrieval, Symbolic Computation, Combined). It also includes a table summarizing relevant datasets. These tables correspond to the synthesis presented in the main Results section.

Table 2: Early Fusion Methods in Vision-Language Model Augmentation. Bracketed tags denote predominant reasoning domain categories: **S**patial, **T**emporal, **K**nowledge grounding, **P**hysical commonsense, **A**ction/Embodiment

| Retrieval | | | |
|---|---|---|---|
| Prompt Augmentation | | Querying KG | Retrieval Encoders |
| Image Caption | Retrieval FT | Prompt Augmentation | Subgraph Enc |
| (Gao et al., 2022a) [**K**] (An et al., 2024) [**K**] (Li et al., 2018) [**S,K**] (Fabian et al., 2023) [**K**] (Sharifymoghaddam et al., 2024) [**K**] (Ghosal et al., 2023) [**K**] (Khademi et al., 2023) [**K**] (Fu et al., 2023) [**K**] (Dey et al., 2021) [**K**] (Lin et al., 2022) [**K**] (Yu et al., 2019) [**S,K**] (an Liu et al., 2024) [**K**] (Mogadala, 2019) [**K**] (Lin & Byrne, 2022) [**K**] (Garcia-Olano et al., 2021) [**K**] (Salemi et al., 2023b) [**K**] (Luo et al., 2021) [**K**] (Vo et al., 2022) [**K**] (Hao et al., 2024b) [**K**] (Gui et al., 2021) [**K**] (Chen et al., 2021a) [**K**] (Liang et al., 2021) [**K**] (Lerner et al., 2023) [**K**] | (Kan et al., 2023) [**K**] (Ranjit et al., 2023) [**K**] (Qu et al., 2024) [**K**] (Liu et al., 2024) [**K**] (Yan & Xie, 2024) [**K**] (Xu et al., 2024a) [**K**] (Khaliq et al., 2024) [**K**] (Xuan et al., 2024) [**K**] (Wen et al., 2024) [**K**] (Iscen et al., 2023) [**K**] (Joshi et al., 2024) [**K**] (Chen et al., 2022b) [**K**] (Gur et al., 2021) [**K**] (Cui et al., 2024) [**K**] (Zhu et al., 2023) [**K**] (Hao et al., 2024a) [**K**] | (Ravi et al., 2022) [**K**] (Narasimhan & Schwing, 2018) [**K**] (Vickers et al., 2021) [**K**] (Guo et al., 2022) [**K**] (Wang et al., 2015) [**S,K**] (Natu et al., 2023) [**K**] (Jhalani et al., 2024) [**K**] (Barezi & Kordjamshidi, 2024) [**K**] (Zhang et al., 2024) [**K**] (Zhang et al., 2023g) [**K**] (Wang et al., 2023) [**K**] (Ogawa et al., 2024) [**T**] (Chen et al., 2022c) [**K**] (Kan et al., 2021) [**S,K**] (Gan et al., 2023) [**K**] (Yang et al., 2019) [**T,K**] | (Li et al., 2020) [**K**] (Rao et al., 2023) [**K**] (Zhang et al., 2020) [**K**] (Lee & Kim, 2021) [**K**] (Li et al., 2022a) [**K**] (Lin et al., 2023a) [**K**] (Torino et al., 2020) [**S,K**] (Wang et al., 2022a) [**K**] (Qu et al., 2020) [**T,K**] (Padhi et al., 2024) [**K**] (Shevchenko et al., 2021) [**K**] (Gardères et al., 2020) [**K**] (Jing et al., 2023) [**K**] (Mondal et al., 2024) [**K**] (Lee et al., 2024) [**K**] |

| Retrieval | | | |
|---|---|---|---|
| Retrieval Encoders (Continued) | | | |
| KG Encoding | | Encoder Architectures | |
| KG Conv | MMKG Attn | Attention | LSTM |
| (Chen et al., 2021b) [**K**] (Ziaeefard & Lécué, 2020) [**K**] (Yu et al., 2020) [**K**] (Zhu et al., 2020b) [**K**] (Hussain et al., 2022) [**K**] (Li & Moens, 2022) [**K**] (Ye et al., 2021) [**S**] | (Jiang & Meng, 2023) [**K**] | (Yuan et al., 2023b) [**K**] (Weng et al., 2024) [**K**] (Chen et al., 2022a) [**K**] (Salemi et al., 2023a) [**K**] | (Wu et al., 2016) [**S,K**] |

| Retrieval | Symbolic |
|---|---|
| Retrieval Encoders (cont.) | |
| Memory | Symbolic |
| (Hu et al., 2022) [**K**] | (Potapov et al., 2019) [**S**] |

Table 3: Middle Fusion Methods in Vision-Language Model Augmentation. Bracketed tags denote predominant reasoning domain categories: **S**patial, **T**emporal, **K**nowledge grounding, **P**hysical commonsense, **A**ction/Embodiment

| Retrieval | | | |
|---|---|---|---|
| Dense Retrieval | Graph | | |
| | KG Prompt Augmentation | KG/NN Similarity | Concept/Scene Fusion |
| (Wang et al., 2022b) [**K**] (Lin et al., 2023b) [**K**] (Jia et al., 2023) [**K**] | (Li et al., 2017) [**K**] (Li et al., 2023b) [**K**] (Zheng et al., 2021) [**K**] (Su et al., 2018) [**K**] (Narasimhan et al., 2018) [**K**] (Zhang et al., 2023c) [**K**] (Singh et al., 2019) [**K**] (Jiang et al., 2020) [**T,A,K**] (Yu et al., 2023) [**K**] (Du et al., 2022) [**K**] (Li et al., 2024a) [**K**] (Yin et al., 2023) [**K**] (Ma et al., 2022) [**K**] (Zhu et al., 2020a) [**S,K**] (Cao et al., 2019) [**S,K**] (Li et al., 2022b) [**K**] (Zheng et al.) [**A,K**] (Wei et al., 2022) [**K**] (Narayanan et al., 2021) [**K**] | (Wu et al., 2024a) [**K**] (Chae & Kim, 2022) [**K**] (Li et al., 2019) [**K**] (ming Xian et al., 2023) [**K**] (Marino et al., 2020) [**K**] | (Yang et al., 2023) [**S,K**] (Wang et al., 2022c) [**S,K**] (Khan et al., 2022b) [**S,K**] (Khan et al., 2022a) [**S,K**] (Zhu, 2022) [**S,K**] (Wen & Peng, 2021) [**S,K**] (Song et al., 2023) [**K,P**] (Li et al., 2022d) [**A,K**] (Zhang et al., 2021) [**S,K**] (Dong et al., 2024) [**S,K**] (Gao et al., 2023a) [**S,A**] (Zhang et al., 2022a) [**K**] (Xu et al., 2021) [**T,K**] (Li et al., 2024b) [**K**] (Hou et al., 2020) [**T,K**] (Gu et al., 2019) [**S,K**] |

| Retrieval | | | Symbolic Computation |
|---|---|---|---|
| Graph | | RL | Program Synthesis |
| MMKGs | Hypergraphs | | |
| (Xi et al., 2024) [**T,K**] (Shi et al., 2022) [**S,K**] (Santiesteban et al., 2024) [**K**] (Ouyang et al., 2024) [**K**] (Liu et al., 2021) [**K**] | (Heo et al., 2022) [**K**] (Wang et al., 2024b) [**K**] | (Bougie et al., 2018) [**A,K**] | (Zhang et al., 2022c) [**K**] (Zhang et al., 2023e) [**S,K**] (Hu et al., 2023b) [**S,K**] (Shirai et al., 2023) [**A,S**] (Zhang et al., 2023b) [**A**] (Li et al., 2021) [**S,K**] (Mishra et al., 2024) [**K**] (Xue et al., 2024) [**S,K**] |

| Symbolic Computation | | | |
|---|---|---|---|
| Logic Engines | VSA | Symbolic Graph Ops | Tool Use |
| (Zhang et al., 2025a) [**K**] (Riley & Sridharan, 2019) [**K**] (Mitchener et al., 2021) [**A**] (Tsatsou et al., 2021) [**K**] (Zhang et al., 2022b) [**A**] (Choi et al., 2024) [**T,K**] (Li et al., 2023c) [**S**] (Li et al., 2023d) [**K**] (Huang et al., 2021) [**K**] (Zhang et al., 2023d) [**A**] | (Montone et al., 2017) [**S**] (Kovalev et al., 2021) [**S**] | (Li et al., 2022c) [**K**] (Liang et al., 2020) [**S,K**] (Wu et al., 2023) [**T,K**] (Zhao, 2015) [**P,S,K**] (Yang et al., 2020) [**S**] (Zhang et al., 2023f) [**K**] (Hudson & Manning, 2019) [**S**] (Cao et al., 2021) [**S,K**] | (Hu et al., 2024) [**S**] (Fan et al., 2024) [**T**] (Liu et al., 2023b) [**A**] (Hu et al., 2023c) [**K**] (Wu et al., 2024b) [**K**] |

| Symbolic Computation | | Combined Retr & Symb |
|---|---|---|
| Self Play | Agents | Other |
| (Misiunas et al., 2024) [**K**] | (Niu et al., 2024) [**A**] (Castrejon et al., 2024) [**K**] (Lu et al., 2023) [**S,K**] (Hsieh et al., 2023) [**K**] (Xu et al., 2024b) [**A,K**] (Yang et al., 2024) [**T**] | (Besbes et al., 2015) [**K**] (Aditya et al., 2016) [**K,P**] (Aditya, 2017) [**S,K**] (Aditya & Baral, 2016) [**K**] (Tan et al., 2021) [**A,K**] (Liu et al., 2023a) [**K**] (Stammer et al., 2024) [**K**] (Vatashsky & Ullman, 2018) [**S,K**] (Gao et al., 2023c) [**S,K**] (Gao et al., 2024) [**S,K**] |

Table 4: Late Fusion Methods in Vision-Language Model Augmentation. Bracketed tags denote predominant reasoning domain categories: **S**patial, **T**emporal, **K**nowledge grounding, **P**hysical commonsense, **A**ction/Embodiment

| Retrieval | | Symbolic Computation | |
| --- | --- | --- | --- |
| Dense | Knowledge Graph | Program Synth | Symbolic Engines |
| (Song et al., 2022a) [**K**] (Song et al., 2022b) [**K**] (Shi et al., 2024) [**S,T,K**] | (Gao et al., 2022b) [**S,K**] (Huang et al., 2020) [**K**] (Xiao & Fu, 2022) [**S,K**] | (Vedantam et al., 2019) [**S**] (Yi et al., 2018) [**S**] (Surís et al., 2023) [**S,P**] (Khandelwal et al., 2023) [**S**] (Subramanian et al., 2023) [**S,K**] (Gupta & Kembhavi, 2022) [**S**] (Bhaisaheb et al., 2023) [**S,K**] | (Sethuraman et al., 2021) [**S,K**] (Aditya et al., 2018) [**S**] (Eiter et al., 2022) [**S,K**] (Eiter et al., 2021) [**K**] (Cunnington et al., 2024) [**K**] |

| Symbolic Computation | | | Combined |
| --- | --- | --- | --- |
| Symbolic Graph Ops | Tool Use | Other | Combined |
| (Li et al., 2023a) [**A,T**] (Zhan et al., 2021) [**S**] (Saqur & Narasimhan, 2020) [**S,K**] (Johnston et al., 2023) [**S,K**] | (Yuan et al., 2023a) [**K**] (Cesista et al., 2024) [**K**] (Cesista, 2024) [**K**] (Zhang, 2023) [**K**] | (Xu et al., 2022) [**A**] (Singh, 2018) [**K**] (Bao et al., 2023) [**K**] (Verheyen et al., 2023) [**T,K**] | (Sachan, 2020) [**K**] (Basu et al., 2020) [**S,K**] |

Table 5: Datasets Relevant to Augmented Vision-Language Models. Bracketed tags denote predominant reasoning domain categories: **S**patial, **T**emporal, **K**nowledge grounding, **P**hysical commonsense, **A**ction/Embodiment

| Spatial Reasoning [S] | | Knowledge Based VQA [K] | Reasoning VQA [S,K] |
| --- | --- | --- | --- |
| CLEVER | Scene Graph | KBVQA | Reasoning VQA |
| (Johnson et al., 2016) (Yi et al., 2019) (Li et al., 2022e) (Abraham et al., 2024) (Wang et al., 2024d) | (Krishna et al., 2016) (Shen et al., 2024) | (Agrawal et al., 2015) (Wang et al., 2016) (Lin et al., 2023c) (Shah et al., 2019) (Marino et al., 2019) (Reichman et al., 2023) (Su et al., 2024) (Mensink et al., 2023) (Jain et al., 2021) (Cao et al., 2020) (Sung et al., 2022) (Agarwal et al., 2024) (Qiu et al., 2024) (Lerner et al., 2022) | (Schwenk et al., 2022) (Gao et al., 2019) (Zellers et al., 2018) (Cao et al., 2019) (Bitton-Guetta et al., 2024) |

| Knowledge and Spatial [K,S] | Agents [A] | Task Specific | |
| --- | --- | --- | --- |
| Knowledge and Spatial | Agents | Robotics [A] | Other (Task) [K] |
| (Chen et al., 2023) (Wang et al., 2024a) (Zhang et al., 2023a) | (Niu et al., 2024) (Zhou et al., 2023) (Cao et al., 2024) | (Gao et al., 2023b) | (Hayashi et al., 2024) (Jin et al., 2024) (Hu et al., 2023a) |

## C   Glossary of Key Terms

Table 6: Key Terms and Definitions

| Term | Definition |
| --- | --- |
| Vision-Language Model (VLM) | Model jointly processing visual and textual modalities through tasks like VQA, captioning, or image-text retrieval. |
| Augmented VLM (AVLM) | VLM integrated with external symbolic systems, APIs, or tools during inference to overcome standalone limitations. |
| Neural-Symbolic System | Hybrid architecture combining neural pattern recognition with symbolic logical reasoning and knowledge representation. |
| Early Fusion | Integration of external information at input stage, before VLM internal processing begins. |
| Middle Fusion | Integration during VLM's inference, interacting with intermediate representations before final output. |
| Late Fusion | Integration after VLM generates initial output, typically for validation, refinement, or explanation. |
| Retrieval Augmented Generation (RAG) | Retrieving relevant external information to provide as context for model generation. |
| Knowledge Graph (KG) | Structured knowledge as directed graph with entity nodes and relationship edges, encoded as triplets. |
| Scene Graph | Structured visual scene representation with object nodes and spatial relationship edges. |
| Program Synthesis | Automatic generation of executable code (Python, SQL) by models for reasoning operations. |
| Tool Use | VLMs dynamically invoking external tools (calculators, APIs, vision modules) based on task needs. |
| Graph Neural Network (GNN) | Neural architecture for graph data, enabling message passing across nodes and edges. |
| Dense Retrieval | A retrieval method using learned dense vector embeddings to find semantically similar content through vector similarity metrics rather than keyword matching. |
| Visual Question Answering (VQA) | Task requiring natural language answers to questions about visual content. |
| ConceptNet | Multilingual common-sense knowledge graph with semantic concept networks. |
| Answer Set Programming (ASP) | Declarative programming paradigm for knowledge representation and logical constraint solving. |
| PDDL | Planning Domain Definition Language - a standardized language for expressing planning problems and domains in automated planning systems. |

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
