# OpenReview forum: "Augmented Vision-Language Models: A Systematic Review"
_TMLR — Accepted by TMLR_

### Review · Reviewer_CEXb · 2025-08-29

**Summary Of Contributions:**

This paper presents a survey of scientific literature on augmented vision language models, i.e., vision-language models that make use of additional functionality beyond the VLM itself, such as prompt augmentation, retrieval, tool calls, or external solvers for generating answers. The paper categorizes augmentation into early-, middle-, and late-fusion and discusses related approaches. The paper concludes with a discussion on current challenges and possible future directions. The supplementary material documents in detail how literature to be included in the survey was collected, and also briefly comments on the limitations of the literature search.

**Audience:**

Yes

**Audience Explanation:**

The main use case of this paper is obtaining a comprehensive overview of the current state of augmented vision language models. This is certainly of interest to a part of TMLR's audience, as vision-language models represent a significant area of current AI/ML research.

**Broader Impact Concerns:**

No broader impact concerns.

**Claims And Evidence:**

Yes

**Claims Explanation:**

This paper presents a survey of existing literature on augmented vision language models, but no empirical results. The methods to search and filter literature for inclusion are documented in the supplementary and follow good practices. The supplementary material also features tables organizing individual publications by keywords, which could serve as a useful resource for researchers seeking an effective reference guide for the field.

**Requested Changes:**

**(R1, critical)** Sections use different citation styles. Section 1 and 7.5 use `\citet`, while Sec. 3 and others `\citep` . In most cases, `\citep` is correct. Please thoroughly review references and adjust them to the context.

**(R2, critical)** Sections 7.1 to 7.4 lack explanations that support the claims. For example, Sec. 7.2 claims "hybrid architectures consistently outperform purely neural methods of similar computational requirements". On which models/papers is this claim based? In Sec. 7.4, it is claimed that "few systematically evaluate both the correctness and interpretability of neural-symbolic systems in a consistent manner". Which are the few approaches that do this? This part is also a great opportunity to highlight promising directions towards unifying evaluation for augmented VLMs, beyond the insight that no generally accepted framework exists at this point? Overall, these sections would significantly benefit from adding explanatory statements and justifications.

**(R3, critical)** Sec. 6 lists many datasets, most of which are not exclusive to augmented VLMs. Here, it is necessary to better highlight the connections of the identified dataset groups to augmented VLMs. How is their use for training or evaluating augmented VLMs different from their use for non-augmented VLMs?

**(R4, critical)** In Sec. 7.7, are there any aspects that are specific to VLMs? Most proposed directions of future work appear to be valid as well for LLMs, for example, better integration with the environment, specialized applications, and human-in-the-loop scenarios. To better align this section with the audience, I recommend adding some specific directions targeting VLMs.

**(R5, non-critical)** The main part of the survey is very concise, mostly explaining important keywords in 1 sentence and then citing relevant literature. I think it would be a really great addition to add a table to the supplementary material containing all keywords listed in Tables 1 to 4 and more detailed definitions of these keywords.

---

> ### Author Response · Authors · 2025-09-16
>
> Thank you for the helpful feedback. We agree with your critiques and have addressed each concern in the latest revision.
>
> R1 - Each citation has been standardized to use \citep.
>
> R2 - This section has been completely rewritten to have more specific claims that are supported by the surveyed papers.
>
> R3 - It is true that most of the previously mentioned datasets were not specific to AVLMs, and so we have removed the section entirely. We have moved the discussion on remaining relevant datasets to Section 6.
>
> R4 - We have rewritten this section to be specific to AVLMs, as the previous version was indeed too general.
>
> R5 - We have added Appendix C as a glossary of key terms.
>
> Thank you again for taking the time to review our submission. We hope the revision has addressed your concerns.

---

### Review · Reviewer_psmb · 2025-09-01

**Summary Of Contributions:**

Vision-Language Models that take in text and image and output text have advanced rapidly in recent years.  Given that they are typically derived from LLMs, VLMs are also undergoing a trend where their shortcomings are addressed by augmenting their abilities with external systems, for example to perform precise math calculations or retrieve external knowledge.

This paper aims to fill a gap in the literature by surveying the state of the art in approaches to augmenting VLMs with external systems.  While previous surveys exist covering specific aspects of augmented VLMs (AVLMs), e.g., knowledge enhancement or the use of neuro-symbolic systems, this can stand as the first systematic study of the full landscape of AVLMs.

## Strengths

Strengths of this paper include:
 - Tool use in LLMs/VLMs is of very high interest to the community
 - The goal of performing the first systematic survey of the full landscape of AVLMs sets the stage for a meaningful contribution.

## Weaknesses

While I agree with the authors that this study has the potential to fill a meaningful gap in the literature, I find that the current version has significant flaws.

First, I found the abstract and introduction to be very confusing given the contents of the paper.  They focus on neurosymbolic approaches to augmenting VLMs, but this is only one aspect of what’s covered in the paper.  For example, the enumeration of the weaknesses of neuro-symbolic approaches seems out of place, especially in the introduction.  The focus here, as I understand it, is instead addressing weaknesses of VLMs.  Such a detailed enumeration of the weaknesses of VLMs, ideally with visual examples, would have been useful early in the paper.  An illustrative Figure 1 would be useful (see below for more thoughts on figures).

Second, my biggest concern is that I find that many citations that I look up do not directly support the sentences or passages in which they are used.  This is a major concern for a survey.  Although I cannot practically review every citation in the paper, here are some examples that I found:

 - In the first sentence of the first paragraph, CLIP is cited to be considered as a VLM.  However, CLIP cannot perform the tasks being discussed, and does not meet the definition of a VLM given in Section 1.2, as it does not generate text.
- Again in paragraph 1, Rudin 2021 and Mitchell 2022 are not strongly related to the points being made.  More specific references should be used, or these references should be positioned alongside statements more accurately reflecting their contents.  Mirzadeh et al 2024 is not about VLMs.  Overall, by my count only 2 of 6 citations from paragraph 1 are about VLMs.  Given the vast body of literature being produced on these models, it is unclear why stronger references cannot be found.
 - I looked at a set of citations from Section 4.2, and found that many did not strongly (or even weakly) support their accompanying passage.  For example, in the sentence about tool use: Hu et al., 2024 is not about  integrating external information or symbolic computation during the VLM’s forward pass, or interaction with the model’s intermediate representations; Fan et al., 2024 is not about tool use; Liu et al., 2023b does not seem to be about middle fusion; Hu et al., 2023c and Wu et al., 2024 are not about LLMs.
 - Suris 2023 does not involve a VLM.  I do not believe that any of its 4 references in the main paper are accurate vis-a-vis what is in this paper.
 - The first sentence of Section 7.6 has two references for AVLMs and tool use,  (Qin et al., 2023; Schick et al., 2023).  Neither of these references feature VLMs (or AVLMs).  I also believe that the Yang and Lu papers referenced in that section do not feature VLMs.
 - Overall, I understand that LLM references are useful here, but the way they are used is very unclear, and usually makes it seem like the reference should be about multimodal models (at least way more often than they are.

Next, I find that the paper lacks depth, such that the paragraphs become highly repetitive and not very informative.  Some problem setup sections (with accompanying visuals) to give basic technical background on VLMs, knowledge graphs, neuro-symbolic systems, tool use, etc. would be very helpful.  As it stands, while I am very familiar with LLMs/VLMs and tool use, after reading this survey I do not have a clear idea of how VLMs interact with KGs or symbolic systems.  More details on some select papers would be helpful in illustrating what is going on here.

I am also confused about how many of these techniques are inference-only; for example, in Section 3.1 (bottom page 5) how does the model condition on retrieved encodings that it has never seen?  How do we incorporate an LSTM as an inference-time technique without training?

I am further unsure of what constitutes an intermediate representation under the conception in this paper, and thus I have trouble understanding how approaches end up in Section 4.  Does a chain of thought constitute an intermediate representation?  Based on the methods included it seems so, but this does not match up with the description at the top of Section 4.

This statement in 7.1 requires citation: “As of the time of writing, many commercially
available AI products use only early fusion retrieval techniques…”

More illustrative figures, especially high-level ones early in the paper, would be very beneficial.  Also, the amount of citations in some sentences make it hard to find the continuation of the sentence.  Some citations might be moved to tables, figures, or elsewhere.

**Audience:**

Yes

**Audience Explanation:**

Yes, a systematic review of AVLMs would be of interest to the TMLR audience.

**Claims And Evidence:**

No

**Claims Explanation:**

I find too many instances where citations do not support the claim that they are referenced in to support this paper’s acceptance in its current form.

**Requested Changes:**

### Important Changes
- Review all citations, make sure they more directly support the accompanying claims.
- Rewrite abstract and introduction to clarify scope; better balance of information about VLMs and different augmentation approaches; less focus on shortcomings of neurosymbolic approaches.
- Give some light technical background, and more depth in places in Sections 3-5, so that a VLM expert could gain a better understanding of how these techniques might work.
- Clarify the criteria for designation as middle vs. late stage fusion.

### Other Suggestions
- More illustrative figures.

---

> ### Author Response · Authors · 2025-09-16
> **Thank you for the review**
>
> We would like to thank you for the helpful feedback. We do agree with your critiques and have uploaded a new revision addressing each concern.
>
> - We have updated the definition of Vision-Language models to more accurately reflect the articles surveyed. Included in our definition is any machine learning model which jointly processes text and images for generative or discriminative tasks. This clears up many of the misaligned citations that you had correctly noted.
> - Each citation was evaluated for relevance to the claim, and additional citations were used to support previously uncited claims, as noted by both reviewers. Many of the problematic sections were rewritten entirely with new citations supporting each claim.
> - The introductory section of neurosymbolic approaches is replaced by a more focused definition and history of augmentation strategies.
> - The discussion section was completely rewritten in order to provide depth to the analysis, replacing repetitive points from the previous revision. Included are more insightful assessments that are specific to augmented vision-language models in the survey, rather than neurosymbolic AI in general.
> - Two additional figures were added early in the review to help communicate the architecture of early and middle fusion methods.
> - An additional statement is appended to the definition of Augmented Model to specify that this survey focuses on inference-time augmentation to distinguish it from training-time dataset augmentations, and that this does not preclude AVLMs from additional finetuning or being inference-only.
> - A clarifying sentence was added regarding the definition of "intermediate representations" in the introduction of Middle Fusion methods to include token-based representations as seen in chains-of-thought.
> - Long chains of citations were replaced with references to the appropriate tables in the Appendix.
>
> Thank you again for your helpful insights, we hope that this revision adequately addresses the noted weaknesses and look forward to additional feedback.

---

### Review · Reviewer_xaWS · 2025-10-01

**Summary Of Contributions:**

This is a survey article about vision-language models that are *augmented*, either with retrieval from a knowledge base or with symbolic computation from an external tool. Unusually for a ML survey, I think, it follows the PRISMA guidelines for systematic reviews of medical studies and gives a detailed record in the appendix of how the surveyed papers were collected. This methodology should also be considered one of the contributions of the paper.

The TMLR guidelines for surveys state that surveys should "draw new connections, highlight trends, and suggest new problems." The categorization of previous approaches into late, middle, and early fusion, and into knowledge augmentation vs. symbolic computation is helpful and can be considered "drawing new connections." The discussion in Section 6 can be considered "highlighting of trends," and the future directions in Section 6.3 can be considered "suggesting new problems."

I feel that the survey is generally dense with citations and doesn't necessarily dig very deeply into them; for example, if there are design decisions that many of the surveyed approaches have in common, it would be worthwhile to present those in greater technical detail. In this respect, the value added by synthesizing all these citations may be low.

Section 6.1 discusses five challenge areas. Unlike Sections 2-5, where the organizational principle is very clear, here I feel uncertain what the organizational principle is and whether this is an exhaustive list of challenge areas. It feels like there might be some principle(s) (e.g., space versus time) that are there, but not made explicit yet.

**Audience:**

Yes

**Audience Explanation:**

As a survey article, it doesn't exactly have findings, but it surveys a topic squarely in the interest area of TMLR's audience. The categorization of these papers, and the discussion of present challenges and some solutions are also clearly in scope for TMLR.

**Claims And Evidence:**

Yes

**Claims Explanation:**

As a survey article, it doesn't exactly have scientific claims, but it promises to systematically survey this area of research, and it does. As mentioned above, it goes to greater lengths than is usual in ML to convince the reader that the survey covers all papers in this area that were available at the time.

**Requested Changes:**

The term "augmented" seems to me to be not very informative (indeed, at first glance, I thought this was a survey on general vision-language models). When I google "augmented vision-language models," the first hit is this paper. Is there not a widely-used term for this area of research?

---

> ### Author Response · Authors · 2025-10-03
>
> Thank you for your review and helpful suggestions.
>
> Firstly, to address the use of the 'augmented' term, it was inspired by a similar survey paper from Yann LeCun's group at Meta titled 'Augmented Language Models: A Survey', published in TMLR in 2023: https://openreview.net/forum?id=jh7wH2AzKK. We find the term appropriate and informative as we extend the research into the vision-language domain.
>
> We have added a deeper analysis into the commonalities of the surveyed approaches in a new section "6.2 Common Architectural Patterns in AVLMs". Three common patterns stand out, and we analyze their evolution and cross-pattern observations.
>
> Lastly, we have categorized each surveyed approach into at least one of 5 reasoning domains: spatial, temporal, knowledge ground, physical commonsense, and action/embodiment. This provides more needed structure to the discussion section, and makes explicit the domain specific limitations that these augmentations address. Also included is a new table which summarizes the cross section of these domains and augmentation techniques.
>
> We hope you find these additions improve the structure and value of the paper.

---

### Decision · Action_Editor_HgHk · 2025-12-18

**Recommendation:** Accept as is

**Additional Comments:**

This is a survey paper with no other contributions.

**Audience:**

Yes

**Audience Explanation:**

VLMs have been prevalent, but augmented VLMs including tool use, external neuro-symbolic models, etc. have not been summarized nicely in the past. This paper proposes terminologies and classifications of those augmented VLMs, and is a reasonable survey of the recent approaches. Although all reviewers have issues of the survey being not particularly deep, it did provide some new categorizations of papers and made some suggestions to future work.

The authors have not set their permissions correctly, leading to the reviewers not being able to see their rebuttals. However, reviewers were able to read the revised version and come to a final conclusion. One reviewers did not consider ViperGPT and related approaches as VLM, the AE looked into the paper and found that ViperGPT used a VLM for matching and generally falls within the realm of augmented VLMs as the authors claimed. Based on the improvements that the paper has made after the initial reviews, the AE decided to recommend acceptance.

**Claims And Evidence:**

Yes

**Claims Explanation:**

The only claim this survey paper makes is that it attempted to categorize the papers, which it did.

---

> ### Author Response · Authors · 2026-01-02
>
> Thank you, the camera ready revision is posted.
>
> We apologize for the confusion on the permissions, we have corrected the permissions and the rebuttals are now visible to everyone.